# A Simple Review of Small Vessel Disease Manifestation in the Brain, Retina, and Kidneys

**DOI:** 10.3390/jcm11195546

**Published:** 2022-09-22

**Authors:** Kinza Abbas, Yezhong Lu, Shreya Bavishi, Nandini Mishra, Saumya TomThundyil, Shreeya Atul Sawant, Shima Shahjouei, Vida Abedi, Ramin Zand

**Affiliations:** 1School of Medicine, Geisinger Commonwealth School of Medicine, Scranton, PA 18510, USA; 2Cell and Molecular Biology Department, Tulane University, New Orleans, LA 70118, USA; 3School of Medicine, Rutgers New Jersey Medical School, Newark, NJ 07103, USA; 4School of Medicine, Rowan University School of Osteopathic Medicine, Stratford, NJ 08084, USA; 5School of Medicine, Midwestern University Chicago College of Osteopathic Medicine, Downers Grove, IL 60515, USA; 6Department of Neurology, Geisinger Neuroscience Institute, Geisinger Health System, Danville, PA 17822, USA; 7Department of Neurology, Barrow Neurological Institute, St. Joseph’s Hospital and Medical Center, Phoenix, AZ 85013, USA; 8Department of Public Health Sciences, College of Medicine, The Pennsylvania State University, Hershey, PA 17033, USA; 9Neuroscience Institute, The Pennsylvania State University, Hershey, PA 17033, USA

**Keywords:** small vessel disease, cerebrovascular disease, stroke, retinopathy, chronic kidney disease, diagnostic imaging, white matter

## Abstract

Small blood vessels express specific phenotypical and functional characteristics throughout the body. Alterations in the microcirculation contribute to many correlated physiological and pathological events in related organs. Factors such as comorbidities and genetics contribute to the complexity of this topic. Small vessel disease primarily affects end organs that receive significant cardiac output, such as the brain, kidney, and retina. Despite the differences in location, concurrent changes are seen in the micro-vasculature of the brain, retina, and kidneys under pathological conditions due to their common histological, functional, and embryological characteristics. While the cardiovascular basis of pathology in association with the brain, retina, or kidneys has been well documented, this is a simple review that uniquely considers the relationship between all three organs and highlights the prevalence of coexisting end organ injuries in an attempt to elucidate connections between the brain, retina, and kidneys, which has the potential to transform diagnostic and therapeutic approaches.

## 1. Introduction

The kidney, brain, and retina are highly metabolic organs that require specialized vascular networks to carry out their function. Cohort studies suggest that retinopathy, cerebrovascular disease (CVD), and chronic kidney disease (CKD) frequently coincide. The relationship could potentially be explained through similar embryological development, small vessel histological structure, and risk factors. In this simple review, we look at current evidence of the prevalence of retinopathy, CVD, and CKD with each other and attempt to elucidate the connection of changes between the organs which could have the potential to transform early diagnostic and therapeutic approaches.
The electronic database PubMed was the primary source for article identification. Articles were searched from database inception to December 2021 and identified through keywords including “cerebrovascular disease”, “chronic kidney disease”, and “retinopathy”. Accessible articles in the English language were appraised and assessed. Articles that included the elements of measured biomarkers of the organs, demographics, and conclusions were considered. 

## 2. Associations between Chronic Kidney Disease and Cerebrovascular Disease in Clinical Studies

Cerebrovascular disease (CVD) often involves cerebral small vessel disease (CSVD) and presents with stroke, cognitive impairment, mood disturbances, and dementia [1]. The imaging markers of CSVD include white matter hyperintensities (WMH), cerebral microbleeds (CMBs), lacunes, cerebral atrophy, and enlarged perivascular spaces (EPVS) [1,2]. The brain and kidneys have similar anatomical and vasoregulatory characteristics, which makes them vulnerable to similar vascular risk factors [1,2]. Declining stages of kidney failure are significantly associated with the presence of CMBs and deep WMHs [3]. Therefore, although the two vascular beds are susceptible to traditional arteriosclerotic factors, cerebrovascular changes and declined kidney function demonstrate correlations not explained by traditional risk factors alone (Table 1).

## 3. Associations between Cerebrovascular Disease and Retinal Changes in Population-Based Studies

The development of digital retinal imaging techniques makes the ability to measure retinal microvasculature possible. Microvascular abnormalities observed through ophthalmoscopes such as vessel caliber, microaneurysm, focal narrowing, and topology could be used as markers for the state of cerebral and renal microvasculature [13,14,15]. Retinal vascular architecture changes and retinopathy are prevalent in acute cerebral vascular events in several cohort studies (Table 2). These results might be indicative of common systemic microvascular pathological development processes. 

## 4. Associations between Kidney Disease and Retina Vasculature Changes in Clinical Studies

Retinopathy and retinal vascular changes have been found to have strong associations with impaired kidney function in multiple cohort studies (Table 3). Retinal arterioles and venules narrow progressively with each stage of renal failure, specifically patients with CKD stages 3 to 5 having narrower retinal microvasculature than patients with CKD 1 and 2 [23]. These findings may suggest shared mechanisms leading to vascular damage in both organs. The associations highlight potential prognostic values using retinopathy to predict the future development of kidney impairments or vice versa.

## 5. Associations between CKD, Retina, and CVD in Basic Science Research

The brain, kidneys, and retina of the eyes are highly vascular organs with similar embryological development, small vessel histological structure, and pathological risk factors. The frequently concurrent prevalence of retinopathy, CVD, and CKD suggests a common basis for pathology.

### 5.1. Embryology and Genetics 

The retina in the optic cup develops from the diencephalon, which is part of the prosencephalon or forebrain. The shared origin and similar development pattern of vasculature by vasculogenesis and angiogenesis may be the cause of linked pathology in the brain and retina [34,35].

The development of the vasculature supplying the brain begins with the formation of six pairs of primitive branchial arch arteries [36,37]. The combination of the third branchial arch arteries and the distal segment of the paired dorsal aortae forms the internal carotid artery, which then branches into anterior and posterior divisions [37]. The anterior division branches into primitive ophthalmic and olfactory arteries, later giving rise to the anterior cerebral, middle cerebral, and anterior choroidal arteries [38]. In contrast, the posterior division gives rise to the posterior cerebral and posterior choroidal arteries [37]. 

The hyaloid artery arises from the primitive dorsal ophthalmic artery and supplies the developing inner retina during its maturation, while the vasculogenesis in the choroid epithelium provides oxygen to the outer avascular retinal layer through diffusion [36,39,40]. The angiogenesis in the retina occurs at the late gestation stage by retinal vasculature, which develops as hyaloid vessels regress [36,39,40,41,42,43]. The vascular layer expands and increases in density following the spread of pigmentation in the retinal epithelium and the production of inductive signals from differentiated retinal pigmental epithelium [44,45,46,47]. 

Similar shared anatomical structures such as fenestrations and regulatory processes, such as the Renin Angiotensin Aldosterone System, may be due to parallel development of glomerular endothelium by vasculogenesis [35]. The retina and kidney also share many developmental pathways, abnormalities, and mutations in molecular pathways, which manifest through features in both organs [48].

Furthermore, cerebral autosomal dominant arteriopathy with subcortical infarcts and leukoencephalopathy (CADASIL) is a rare genetic disorder that primarily affects small vessels in the brain with renal impairment, suggesting a common pathogenetic mechanism of renal and brain lesions in this disease [49]. Retinal axonal loss in CADASIL leads to thinning of the retinal nerve fiber length, retinal vein occlusion, macular edema, and retinal venous disease have also been reported among CADASIL patients [50]. Another autosomal dominant genetic disorder that links these organs is retinal vasculopathy with cerebral leukodystrophy and systemic manifestations (RVCL-S), which presents with cerebroretinal vasculopathy (CRV), WMHs, hereditary vascular retinopathy (HVR), and hereditary endotheliopathy with retinopathy, nephropathy, and stroke (HERNS) [51]. In addition, the growing evidence for the genetic basis of SVD, stemming from MRI studies of WMH, estimate heritability to range between 55–75% [51]. Monogenic SVDs, such as COL4A1/A2-related angiopathies and retinal vasculopathy with cerebral leukodystrophy (RVCL), demonstrate cerebral, ocular, and renal abnormalities, implying that patients with monogenic CSVD have concomitant retinopathy and kidney damage [52]. These provide further indication of shared genetic and developmental similarities in the brain, kidney, and retina. 

### 5.2. Cardiovascular Physiology

The frictional force of blood in the vessels, otherwise known as wall shear stress, is counteracted by tension and stretch in the vessel wall. These strains and stretch forces exert a vaso-protective role by providing hemostatic balance. Ito et al. proposed the “strain vessel hypothesis” that detailed the concept that high tone vessels evolutionarily developed to maintain perfusion of the brain and kidney. Although they are exposed to high blood pressure, vascular damage from prolonged hemodynamic stress and a perturbation to this biochemical balance results in either physiological adaptation or disease of the vessel walls [1,53,54]. Thus, common vascular risk factors such as high blood pressure and diabetes mellitus may cause damage to these strain vessels in parallel in the brain and kidney [54]. Non-traditional CKD-related risk factors, such as chronic inflammation, endothelial dysfunction, and uremic toxins, can promote cerebrovascular injury by triggering vascular injury and endothelial dysfunction [55].

### 5.3. Pathology

The numerous anatomical and functional similarities in the brain, retina, and kidney provide a basis for common pathology. The blood–brain barrier is formed by endothelial cells of the capillary wall with tight junctions, astrocytes with projections sheathing the capillary, and pericytes embedded in the capillary basement membrane [56]. Like the brain, the retina possesses barrier circulation; the blood–retina barrier is arranged as inner and outer barriers [57]. The inner retinal barrier is comprised of endothelial cells with tight junctions and is surrounded by glial cells to maintain retinal homeostasis [57,58]. 

Similar to the glomerular filtration barrier, the outer retinal barrier formed at the retinal pigment epithelial cell layers functions to regulate the movement of solutes and nutrients from the choroid to the sub-retinal space [57,59]. The arrangement of the choriocapillaris, Bruch’s membrane, and retinal pigment epithelial interface are homologous to the endothelium, glomerular basement membrane, and podocytes in the glomerulus (Figure 1) [60]. Physiologically, the glomerulus and choriocapillaris are both fenestrated, while the podocytes and retinal pigment epithelial cells both function as metabolic barriers, actively mediating the exchange of molecules [59,61]. Additionally, Bruch’s membrane and the glomerular basement membrane both contain a network of a3, a4, and a5 type IV collagen chains [59,62]. 

The anatomical and compositional similarities manifest as simultaneous retinopathy and nephropathy in membranoproliferative glomerulonephritis type II and Alport syndrome [63,64,65,66]. The simultaneous pathology and appearance of clinical symptoms in chronic cardiovascular diseases, hypertension, and diabetes mellitus further provide evidence of a link between these organs. Hemodynamic similarities of vascular beds of the brain and kidney are generally noted as the shared mechanism linking cerebral and renal vascular dysfunction biomarkers. The kidney and brain are uniquely low-resistance end-organs exposed to consistent high-volume blood flow [54]. 

Thus, despite differences in location and responses to injury, concurrent changes are seen in the micro-vasculature of these organs under pathological conditions, which we hypothesize to be due to their common embryological, functional, histological, and genetic basis (Figure 2). While the cardiovascular basis of pathology in association with the brain, kidney, and retina is well documented, it is well beyond the scope of this review which is focused on direct pathological associations between them. 

## 6. Clinical Implications and Future Direction

This review illustrates the association of synchronous vascular dysfunction in the brain, retina, and kidneys. The concurrent impairments seen within the three organs are supported by anatomical, physiological, and histological similarities in their microvascular circulations. This relationship is further emphasized in certain disease processes, such as in patients with diabetes mellitus, which involves microvasculature in the brain, kidney, and retina. 

Currently, there is an abundance of cross-sectional studies showing associations of concurrent impairment of the brain, retina, and kidneys, demonstrating major impacts on patients’ quality of life and burdens to the health care system. Identifying the early signs of disease or circulating biomarkers could redefine therapeutic windows, while biomarkers could potentially stratify high-risk populations and provide a cost-effective method to monitor health interventions. For instance, the visualization of the retinal microvasculature in a routine health examination can provide a unique window to study the systemic microvasculature and predict the occurrence of CKD, stroke, and CSVD. Aronov and colleagues’ recent systematic review suggested that the retinal microvasculature can provide essential data about concurrent kidney disease status and predict future risk for kidney disease development and progression [67]. They further observed that patients whose retinal photographs were ungradable manifested the strongest associations with the incidence of end-stage renal disease [67]. 

Similarly, a meta-analysis on the prediction of stroke based on retinal vessel caliber utilizing individual data records on 20,798 middle to older-aged individuals without diabetes indicated that wider retinal venular caliber is independently associated with an increased risk of stroke events [68]. Studies also suggested that the retinal microvasculature can provide essential data about concurrent cardiac disease status and predict future risk of cardiac-related events [69]. 

This is a simple review article that explores the relationship between the brain, retina, and kidneys. Although we recognize additional literature is present, our aim was to bring forward the existence of a relationship between the three organs. Further, we believe that the robust amount of literature warrants further exploration at the very least. The viability of utilizing the discussed biomarkers to screen pathologies in other organs is unclear. Prospective studies aimed at recognizing such common biomarkers may aid in predicting the susceptibility of injury in the other organs. When combined with clinical, genomics, and imaging data, this data can play an important role in precision medicine and predicting outcomes. 

## Figures and Tables

**Figure 1 jcm-11-05546-f001:**
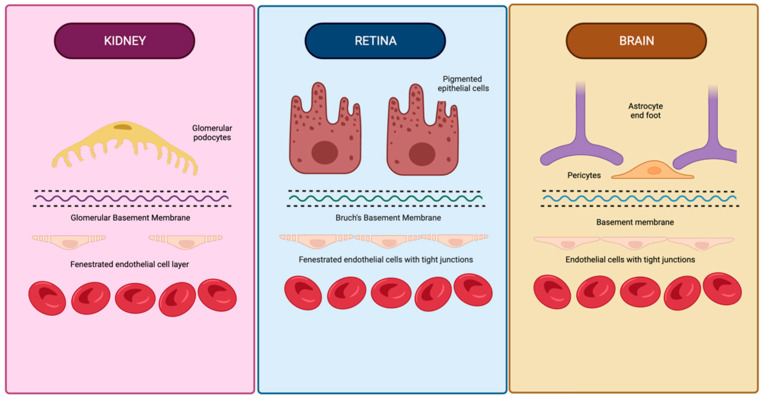
Similarities in anatomical structure of vascular barriers in the brain, retina, and kidneys. The vascular barriers of the brain, retina, and kidneys have many structural similarities, including basement membranes containing type IV collagen, an outer layer with analogous projections (glomerular podocyte foot processes, retinal epithelial cell projections, and astrocyte foot processes), and a fenestrated endothelial cell layer in the glomerulus and retina. Created using BioRender.com on 26 August 2022.

**Figure 2 jcm-11-05546-f002:**
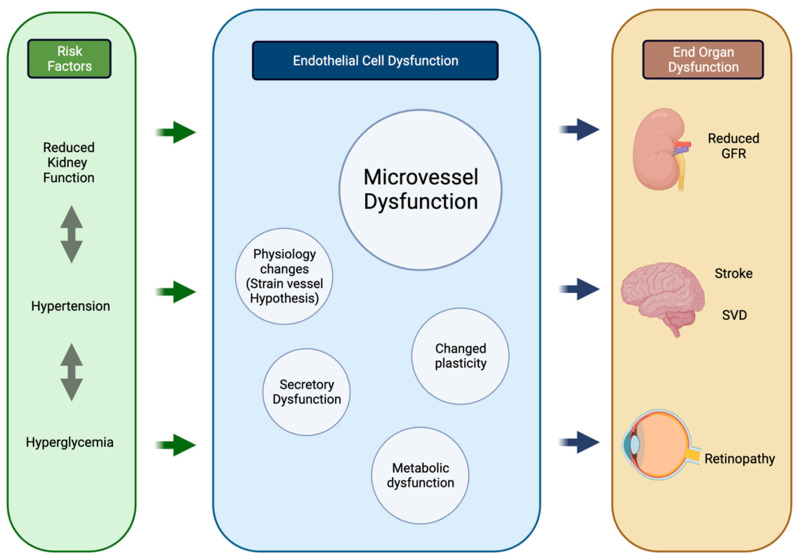
Process of parallel onset end-organ dysfunction. Flowchart demonstrating the interactions between shared systemic risk factors, resulting in endothelial cell changes and end-organ manifestations in the kidneys, brain, and retina. Created using BioRender.com. GFR glomerular filtration rate, SVD small vessel disease. Created using BioRender.com on 25 August 2022.

**Table 1 jcm-11-05546-t001:** Association of kidney function and cerebral vascular changes from population-based studies.

Outcome	Study	Type	Sample Size	Demographics	Conclusion
Stroke/hemorrhage and GFR	Ell Husseini et al. (2018) [2]	Prospective cohort	204,652	Age: >65 years	Within a year after hospitalization for ischemic stroke, GFR and dialysis status on admission are associated with post-stroke mortality (HR 2.09, 95% CI 1.66–2.63) and hospital readmissions (HR 2.55, 95% CI 2.44–2.66).
	Lee et al. (2010) [3]	Meta-analysis	284,672	Varying	A baseline GFR <60 mL/min/1.73 m^2^ was independently related to incident stroke (RR 1.43, 95% CI 1.31 to 1.57; *p* < 0.001).
	Molshatzki et al. (2011) [4]	Prospective cohort	128	Avg age: 71.7 years	Patients with moderate/severe CKD (GFR <45) had >4-fold adjusted hazard ratio for mortality over 1 year (4.29; 95% CI = 1.69–10.90) and a 2.3-fold higher hematoma volume (*p* = 0.04) compared to patients with no impairment.
	Molnar et al. (2016) [5]	Retrospective cohort	516,197	Age: ≥40 years	Incidence of hemorrhage increased 20-fold across declining eGFR and increasing urine ACR groupings (highest eGFR/lowest ACR: 0.5%; lowest eGFR/highest ACR: 10.1%).
CMBs and GFR	Kim et al. (2017) [6]	Cross-sectional	2518	Age: 40–79 years without symptomatic stroke history	Subjects with CMB demonstrated a higher proportion of moderate-to-severe renal dysfunction than those without CMB (15.5% vs. 5.0%, *p* < 0.001).
	Ovbiagele et al. (2013) [7]	Retrospective/prospective cohort	197	Age: ≥18 years with primary ICH	CKD was associated with the presence of CMB (adjusted OR, 2.70; 95% CI, 1.10–6.59) and number of CMB (adjusted RR, 2.04; 95% CI, 1.27–3.27.
	Cho et al. (2009) [8]	Retrospective cohort	142	Age: Avg 66.7 years	Low GFR levels were associated with the presence of cerebral microbleeds (OR, 3.85; 95% CI, 1.52 to 9.76, *p* = 0.004).
WMHs and kidney function	Akoudad et al. (2015) [9]	Prospective cohort	2526	Age: ≥45 years	Worse kidney function was consistently associated with a larger white matter lesion volume (mean difference per standard deviation increase in albumin-to-creatinine ratio: 0.09, 95% CI 0.05; 0.12; per standard deviation decrease in creatinine-based GFR: −0.04, 95% CI −0.08;−0.01).
	Kim et al. (2019) [10]	Cross-sectional	2203	Age: ≥40 years	Subjects with both significant albuminuria and GFR < 60 had a significantly higher WMH volume (β = 0.652; *p* < 0.001).
	Wada et al. (2008) [11]	Cross-sectional	625	Age: 61–72 years	Subjects with lower GFR levels tended to have more lacunar infarcts and higher grades of WMHs. In addition, the mean grades of WMHs or the mean number of lacunar infarcts in the subjects with albuminuria were greater than those in subjects without albuminuria.
	Xiao et al. (2015) [12]	Cross-sectional	413	Age: Avg 64 years	Proteinuria and impaired eGFR were correlated with the severity of EPVS in both centrum semiovale (OR 2.59; 95% CI 1.19–5.64 and OR 2.37; 95% CI 1.19–4.73) and basal ganglia (OR 5.12; 95% CI 2.70–12.10 and OR 4.17; 95% CI 2.08–8.37).

GFR glomerular filtration rate, CMB cerebral microbleeds, CKD chronic kidney disease, CI confidence interval, WMH white matter hyperintensity, OR odds ratio, RR relative risk. Note: We recognize that additional literature is available; this table includes a sample of relevant articles and findings.

**Table 2 jcm-11-05546-t002:** Association of retinopathy and retinal vascular changes in cerebral vascular disease from population-based studies.

Outcome	Study	Type	Sample Size	Demographics	Conclusion
White Matter Disease and retinopathy	Wong et al. (2002) [16]	Prospective cohort	1684	Age: 51–72 years	Persons with retinopathy were more likely to have WMLs than those without retinopathy (22.9% vs. 9.9%; OR, 2.5; 95% CI: 1.5–4.0).
Stroke and retinopathy	Wong et al. (2002) [16]	Cross-sectional	2050	Age: 69–97 yearsWithout diabetes	Retinopathy was found to be associated with prevalent stroke (OR 2.0).
	Cooper et al. (2005) [17]	Cross-sectional	1684	Age: 55–74 yearswithout a history of clinical stroke	Cerebral infarcts were found associated with soft exudates (OR 2.08; 95% CI: 0.69–6.31).
	Mitchell et al. (2005) [18]	Prospective cohort	3654	Age: >49 years	Retinopathy was significantly associated with combined stroke events (RR 1.7; 95% CI: 1.0–2.8) in persons without diabetes.
Stroke and retinal vascular changes	Longstreth et al. (2006) [19]	Case-control	1717	Age: >65 years	Smaller arteriovenous ratio (per standard deviation decrease) was found associated with prevalent infarcts (OR 1.18; 95% CI: 1.05–1.34; *p* = 0.007). Arteriovenous nicking was found associated with prevalent (OR 1.84; 95% CI, 1.23, 2.76; *p* = 0.003) and incident (OR 1.84; 95% CI: 1.15–2.94; *p* = 0.011) infarcts.
	Wong et al. (2006) [20]	Prospective cohort	1992	Age: 69–97 years	Larger retinal venular caliber was associated with incident stroke (OR 2.2; 95% CI: 1.1–4.3).
	Ikram et al. (2006) [21]	Prospective cohort	5540	Age: >55 Without history of stroke	Larger venular diameters were associated with an increased risk of stroke (HR per SD increase 1.12; 95% CI: 1.02–1.24) and cerebral infarction (HR: 1.15; 95% CI: 1.02–1.29).
	Mitchell et al. (2005) [22]	Prospective cohort	3654	Age: >49 years	Combined stroke events were more frequent in participants with retinopathy (5.7%), with moderate/severe arteriovenous nicking (4.2%), or with focal arteriolar narrowing (7.2%) compared with those without (1.9%).

WML white matter lesion, OR odds ratio, CI confidence interval, RR relative risk, HR hazard ratio. Note: We recognize that additional literature is available; this table includes a sample of relevant articles and findings.

**Table 3 jcm-11-05546-t003:** Association of retinopathy and retinal vascular changes with CKD from population-based studies.

Outcome	Study	Type	Sample Size	Demographics	Conclusion
CKD and retinopathy	Grunwald et al. (2019) [24]	Case-Control	1025	Participants with CKD	CKD progression associated with worsening of retinopathy in comparison with participants with stable retinopathy (OR 2.24; 95% CI: 1.28–3.91).
	Deva et al. (2011) [25]	Case-Control	150	Participants with CKD stage 3–5 Australia	CKD stages 3 to 5 (OR 1.79; CI: 1.00–3.20) were independent determinants of macular degeneration.
	Gao et al. (2011) [26]	Cross-sectional	9670	Participants with CKD	Prevalence of retinopathy is higher in CKD patients than participants without CKD (28.5% vs. 16.3%, *p* < 0.001).
	Choi et al. (2011) [27]	Cross-sectional	3008	Age: 50–87 Participants with CKD South Korea	Participants with CKD are more likely to develop early age-related macular degeneration (OR 1.68; 95% CI: 1.04–2.72) and peripheral retinal drusen (OR 2.01; 95% CI: 1.02–3.99) than those without.
	Liew et al. (2008) [28]	Case-Control	1183	Age: >54 Participants with CKD Australia	Individuals with moderate CKD were three times more likely to develop early AMD than individuals with no/mild CKD (OR 3.2; 95% CI: 1.8–5.7).
CKD and retinal vascular/Structural changes	Balmforth et al. (2016) [29]	Prospective cross-sectional	150	50 patients with hypertension (clinic BP greater than or equal to 140/90 mmHg prior any treatment) 50 with CKD 50 matched healthy controls	Retinal thickness, macular volume, and choroidal thickness were all reduced in CKD compared with hypertensive and healthy subjects.
	Liew et al. (2012) [30]	Cross-sectional	2971	Age: >49	CKD was associated with both presence of retinopathy (OR, 1.2, 95% CI: 1.0–1.5) and venular dilation (OR 1.2, 95% CI: 1.0–1.5).
	Awua-Larbi et al. (2011) [31]	Case-Control Study	675	Age: 45–84 Without baseline clinical cardiovascular disease	Both narrower CRAE (OR 1.55; 95% CI: 1.17–2.04) and wider CRAE (OR 1.44; 95% CI: 1.07–1.93) were significantly associated with albuminuria.
	Sabanayagam et al. (2009) [32]	Cross-sectional study	2380	Age: 40–80 Singapore	CRAE was associated with CKD, (OR 1.42; 95% CI: 1.03–1.96) for eGFR < 60 and 1.80 (1.11–1.96) for micro/macroalbuminuria.
	Wong et al. (2004) [33]	Prospective study	10,056	Age: 45–64	Individuals with retinopathy (OR 2.0; 95% CI: 1.4–2.8), microaneurysms (OR, 2.0; 95% CI: 1.3–3.1), retinal hemorrhages (OR, 2.6; 95% CI: 1.6–4.0), soft exudates (OR, 2.7; 95% CI: 1.6–4.8), and arteriovenous nicking (OR, 1.4; 95% CI: 1.0–1.9) were more likely to develop renal dysfunction than individuals without these abnormalities.

CKD chronic kidney disease, OR odds ratio, CI confidence interval, AMD age–related macular degeneration, CRAE Central Retinal Arteriolar Equivalent. Note: We recognize that additional literature is available; this table includes a sample of relevant articles and findings.

## Data Availability

Not applicable.

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
