# Peer review of "A Simple Review of Small Vessel Disease Manifestation in the Brain, Retina, and Kidneys"

_jcm, 2022, doi:10.3390/jcm11195546_

Round 1

Reviewer 1 Report

1. Abstract and keywords are needed

2. Figures on pathology part are needed

3. Figure explaining connections/associations and possible pathophysiology between these organs are needed

4. Discussions on different CKD staging and different types of small vessel disease are encouraged

5. Table are not completed, for example "Cho et al. (2009)" Age "???"

Author Response

We thank the reviewer for their time and contributions.

  1. Abstract and keywords are needed  

Thank you for the comment. Abstract and keywords have been added to their respective locations.

  1. Figures on pathology part are needed 

Thank you for the suggestion. The pathology section discusses the similarity in vascular layers within the three organs, which we thought was a concept that may benefit from a visual depiction, therefore a figure of it has been added to the end of the Pathology section.

  1. Figures explaining connection/ associations and possible pathophysiology between these organs are needed 

Thank you for the comment, a figure illustrating possible pathophysiology between these organs has been added after the second paragraph of the Pathology section.

  1. Discussions on different CKD staging and different types of small vessel disease are encouraged 

Thank you for the comment. Studies exploring the stages of CKD staging and small vessel disease in the eye and brain were included in the tables, but we have specifically commented on this relationship in the first paragraph of the “Associations between kidney disease and retina vasculature changes in clinical studies” (fourth line) and “Associations between chronic kidney disease and cerebrovascular disease in clinical studies” (second line). 

  1. Table are not completed, for example "Cho et al. (2009)" Age "???" 

Thank you for the comment. Tables have been reviewed again and missing age has been added to the table. 

Reviewer 2 Report

The authors have summarized associations found between CKD, retinopathy and cerebrovascular diseases using population-based studies. They have discussed thereafter the possible link between these pathologies based on anatomical and functional vascular similarities. They conclude on the need of prospective studies and discuss the perspectives of combining different types of biomarkers for earlier diagnosis and treatment. 

This review is overall useful as it reminds the vascular similarities between the 3 vascular beds and the possible subsequent pathological complications arising from genetic or cardiovascular complications. It can help to promote transdisciplinary research to validate biomarkers common for SVD manifestations.

I have listed several suggestions that the authors should consider to improve the manuscript:

-  In the abstract and introduction/background part, the authors should clarify the aim of their review and highlight the added value compared to previously published meta-analyzes or reviews. The information in the sentence page 8 could be used for this purpose “While the cardiovascular basis of pathology in association with the brain, kidney, and retina is well documented, it is well beyond the scope of this review which is focused on direct pathological associations between them”. 

-     The authors should explain the method used to identify the cohort studies described in Tables 1 , 2 and 3 – It is unclear how the studies were identified (keywords, databases), selected or rejected and what were the inclusion/exclusion criteria used. Were there any quality verifications performed?

-       P2, the authors referred to the “strain vessel hypothesis”. It would add clarity for the readers not in the field to explain this concept further.

-       Paragraph “embryology”: the authors have focused their discussion on the shared embryological development of the retinal and cerebral vasculature. The reader would also expect a similar parallel with the renal vasculature. Adding some elements on it would make the discussion more coherent. The paragraph could also end on a concluding statement;

-       Paragraph “genetics”: this part could be further extended by adding other examples and expanding on the description of other monogenic SVDs;

-    Abstract appears  in “Introduction” section instead; keywords are missing too.

Author Response

  1. In the abstract and introduction/background part, the authors should clarify the aim of their review and highlight the added value compared to previously published meta-analyzes or reviews. The information in the sentence page 8 could be used for this purpose “While the cardiovascular basis of pathology in association with the brain, kidney, and retina is well documented, it is well beyond the scope of this review which is focused on direct pathological associations between them”.  

Thank you for the comment. We agree and have clarified this point in our abstract and introduction.

  1. The authors should explain the method used to identify the cohort studies described in Tables 1 , 2 and 3 – It is unclear how the studies were identified (keywords, databases), selected or rejected and what were the inclusion/exclusion criteria used. Were there any quality verifications performed? 

Thank you for the suggestion. We have detailed our method at the end of our introduction paragraph.

  1. P2, the authors referred to the “strain vessel hypothesis”. It would add clarity for the readers not in the field to explain this concept further. 

Thank you for your comment. Our discussion on the strain vessel hypothesis has been deleted from the "Associations between chronic kidney disease and cerebrovascular disease in clinical studies" paragraph and has solely been expanded on under the Cardiovascular Physiology section in attempts to reduce any confusion and focus our explanation of the concept in one location of the manuscript.

  1. Paragraph “embryology”: the authors have focused their discussion on the shared embryological development of the retinal and cerebral vasculature. The reader would also expect a similar parallel with the renal vasculature. Adding some elements on it would make the discussion more coherent. The paragraph could also end on a concluding statement; 

Thank you for the comment. We have included an additional paragraph (third paragraph) under the Embryology section to address this point. We have also combined the Embryology and Genetics sections to address both comments and help make the discussion of these topics more coherent and intertwined.

  1. Paragraph “genetics”: this part could be further extended by adding other examples and expanding on the description of other monogenic SVDs; 

Thank you for the comment. Genetics section has been combined with Embryology to be more coherent. Details of other monogenic SVDs have been added. 

  1. Abstract appears in “Introduction” section instead; keywords are missing too. 

Thank you for the comment. Abstract and keywords have been added to their respective locations.

Round 2

Reviewer 1 Report

Abbreviations should also be provided for all abbreviations listed in the tables and figures 

Author Response

We thank the reviewer for their time and contributions.

  1. Abbreviations should also be provided for all abbreviations listed in the tables and figures

Thank you for the comment. Abbreviations have been explained in the caption of Figure 2.